# Descriptive Analysis of Trauma Admission Trends before and during the COVID-19 Pandemic

**DOI:** 10.3390/jcm13010259

**Published:** 2024-01-02

**Authors:** Mariusz Jojczuk, Katarzyna Naylor, Adrianna Serwin, Iwona Dolliver, Dariusz Głuchowski, Jakub Gajewski, Robert Karpiński, Przemysław Krakowski, Kamil Torres, Adam Nogalski, Ahmed M. Al-Wathinani, Krzysztof Goniewicz

**Affiliations:** 1Department of Trauma Surgery and Emergency Medicine, Medical University of Lublin, 20-081 Lublin, Poland; serwinadrianna96@gmail.com (A.S.); iwonadolliver@gmail.com (I.D.); adam.nogalski@umlub.pl (A.N.); 2Independent Unit of Emergency Medical Services and Specialist Emergency, Medical University of Lublin, Chodzki 7, 20-093 Lublin, Poland; katarzyna.naylor@umlub.pl; 3Department of Health Promotion, Faculty of Health Sciences, Medical University of Lublin, Staszica 4/6, 20-081 Lublin, Poland; 4Department of Computer Science, Faculty of Electrical Engineering and Computer Science, Lublin University of Technology, Nadbystrzycka 38A, 20-618 Lublin, Poland; d.gluchowski@pollub.pl; 5Department of Machine Design and Mechatronics, Faculty of Mechanical Engineering, Lublin University of Technology, Nadbystrzycka 36, 20-618 Lublin, Poland; j.gajewski@pollub.pl (J.G.); r.karpinski@pollub.pl (R.K.); 6Orthopedics and Sports Traumatology Department, Carolina Medical Center, Pory 78, 02-757 Warsaw, Poland; 7Department of Didactics and Medical Simulation, Medical University of Lublin, Chodzki 7, 20-093 Lubln, Poland; kamil.torres@umlub.pl; 8Department of Emergency Medical Services, Prince Sultan bin Abdulaziz College for Emergency Medical Services, King Saud University, Riyadh 11451, Saudi Arabia; 9Department of Security, Polish Air Force University, 08-521 Deblin, Poland; k.goniewicz@law.mil.pl

**Keywords:** traumatic injuries, hospitalization, COVID-19 pandemic, rural and urban healthcare disparities, ICD-10 classification, healthcare systems resilience, gender disparities in trauma, public health crises

## Abstract

Introduction: Traumatic injuries are a significant global health concern, with profound medical and socioeconomic impacts. This study explores the patterns of trauma-related hospitalizations in the Lublin Province of Poland, with a particular focus on the periods before and during the COVID-19 pandemic. Aim of the Study: The primary aim of this research was to assess the trends in trauma admissions, the average length of hospital stays, and mortality rates associated with different types of injuries, comparing urban and rural settings over two distinct time periods: 2018–2019 and 2020–2021. Methods: This descriptive study analyzed trauma admission data from 35 hospitals in the Lublin Province, as recorded in the National General Hospital Morbidity Study (NGHMS). Patients were classified based on the International Classification of Diseases Revision 10 (ICD-10) codes. The data were compared for two periods: an 11-week span during the initial COVID-19 lockdown in 2020 and the equivalent period in 2019. Results: The study found a decrease in overall trauma admissions during the pandemic years (11,394 in 2020–2021 compared to 17,773 in 2018–2019). Notably, the average length of hospitalization increased during the pandemic, especially in rural areas (from 3.5 days in 2018–2019 to 5.5 days in 2020–2021 for head injuries). Male patients predominantly suffered from trauma, with a notable rise in female admissions for abdominal injuries during the pandemic. The maximal hospitalization days were higher in rural areas for head and neck injuries during the pandemic. Conclusions: The study highlights significant disparities in trauma care between urban and rural areas and between the pre-pandemic and pandemic periods. It underscores the need for healthcare systems to adapt to changing circumstances, particularly in rural settings, and calls for targeted strategies to address the specific challenges faced in trauma care during public health crises.

## 1. Introduction

Injuries stand as a crucial global health concern, extending beyond their immediate medical implications to profound socioeconomic effects. Trauma, according to various sources, accounts for about 6–9% of all deaths worldwide [1,2], with over 5 million fatalities annually. The World Health Organization reports that traumatic injuries make up to 9% of global mortality causes [3]. In Poland, particularly in the Lubelskie Voivodship, traumatic injuries mirror this global trend, presenting unique challenges to the local healthcare system and underscoring the need for region-specific studies in trauma care. Particularly among individuals aged 1 to 44, injuries are a leading cause of death, creating significant financial burdens due to treatment costs, work absences, and long-term treatments. Notably, the consequences like disability impact the quality of life profoundly [4,5,6,7,8].

The Lubelskie Voivodship, located in eastern Poland, faced unique challenges during the COVID-19 pandemic. The region’s response to the pandemic included varying degrees of lockdown and public health policies, which had a profound impact on the population’s mobility and healthcare access. These policies, in tandem with the socio-demographic and occupational profile of the province, likely influenced the patterns of trauma admissions observed in our study.

The Lubelskie Voivodship is characterized by a diverse socio-demographic landscape. It has a mix of urban and rural populations, with significant portions of the workforce engaged in agriculture, industry, and services. The province’s demographic profile, with a substantial proportion of young and middle-aged adults, combined with the nature of predominant occupations, might have influenced both the incidence and types of trauma injuries reported.

Survivors of severe trauma often face considerable challenges, including physical impairments, chronic pain, and mental health issues like post-traumatic stress disorder. Factors contributing to post-traumatic social dysfunction include low educational levels, extended hospitalization, and a history of psychiatric disorders [5]. Education level, in particular, is strongly linked to the long-term outcomes in trauma patients [7].

The Centers for Disease Control and Prevention (CDC) data shows that in the United States, injuries are the third leading cause of death, following cardiovascular and neoplastic diseases [2]. They are the primary mortality cause in the 1–44 age group, with a higher incidence in men [1]. In 2014, unintended traumatic injuries led to 136,053 deaths in the U.S., with an additional 29 million injuries requiring emergency department visits [9].

Elderly patients are also significantly impacted by traumatic injuries, often experiencing worse outcomes due to comorbidities. Factors like advanced age, male gender, existing conditions, medications, and the injury’s severity and mechanism are critical prognostic elements [9,10,11]. The mortality rate among patients over 65 is notably higher, sometimes up to four times that of younger patients [12,13].

Brain injuries and hemorrhages are among the most common causes of death following traumatic injuries, often leading to immediate or rapid post-hospitalization fatalities. Moore et al. suggest that a significant portion of those who survive the initial injury succumb within the first 48 h [9]. Injury prevention is thus a crucial strategy for mortality control, involving societal understanding, behavior change, and engagement in safety practices.

The National Academy of Medicine emphasizes stopping the injury-causing processes as the most effective way to prevent preventable deaths [9]. The economic impact is enormous, with costs in medical care and productivity loss reaching $670 billion in 2013. Preventable injury-caused deaths often involve factors like alcohol, narcotics, lack of safety devices, and high speed [9]. Despite prevention efforts, traffic accidents remain a leading cause of unintentional death, especially among the young.

The societal costs of injuries are highlighted by measures such as Disability Adjusted Life Years (DALY) and Potential Years of Life Lost (PYLL). The Global Burden of Disease study indicates that since 2013, over 900 million people have been affected by injuries, with 4.8 million fatalities [14]. Traumatic injuries are responsible for 250 million DALYs [15], with traffic accidents being a significant contributor to lost healthy life years [16,17].

The integration of artificial intelligence in medicine is gaining recognition, particularly in diagnostics and injury outcome prediction [18,19,20,21,22,23,24,25]. The efficacy of neural networks in medical applications was first demonstrated by Penny and Frost in 1996 [26], showing potential comparable to clinical reasoning. Recent efforts to apply AI to injury outcome assessment using the International Classification of Diseases highlight the need for nonlinear predictive models [27,28,29,30].

During the COVID-19 pandemic, the healthcare system in Lublin, like many others, underwent significant adaptations [31]. These changes included reallocation of resources, shifts in trauma care priorities, and adaptations in healthcare delivery to meet the challenges posed by the pandemic. This evolving healthcare landscape in the region inevitably impacted the patterns of trauma care and hospitalization.

This evolving healthcare landscape in the region inevitably impacted the patterns of trauma care and hospitalization. The pandemic-induced changes in daily life, including extended periods of home confinement, altered work environments, and reduced outdoor activities, contributed to shifts in the nature and frequency of trauma injuries [32]. Specifically, these societal changes may have played a role in the observed increase in abdominal injuries among women, potentially linked to increased domestic responsibilities and altered household dynamics during lockdowns. Additionally, the psychological stress and changes in mental health brought on by the pandemic may have influenced injury patterns, underlining the complex interplay between societal factors and health outcomes.

Despite the clear impact, there remains a paucity of research focused on the specific effects of the pandemic on trauma patterns and hospitalization trends in the region, a gap that our study aims to address.

This study aims to assess the predictive value of the International Classification of Diseases 10th Revision (ICD-10) Classification and neural networks in diagnosing mortality risk in injury patients from the Lubelskie Voivodship in Poland between 2019 and 2021. It seeks to enhance our understanding of trauma outcomes, informing both care and preventive strategies.

## 2. Materials and Methods

### 2.1. Data Collection

This study analyzed trauma admissions from 35 hospitals in Lublin Province, Poland, using data from the National General Hospital Morbidity Study (NGHMS). The NGHMS, a database by the National Institute of Public Health-National Institute of Hygiene (NIPH-NIH), records hospital morbidity across Poland, requiring hospitals to report inpatient discharges with relevant ICD-10 codes. Our focus was on severe traumatic injuries under ICD-10 Chapter XIX. We compared data from an 11-week period during the initial 2020 COVID-19 lockdown (March 12th to May 30th) with the same timeframe in 2019. This duration was chosen to observe the immediate impact of the pandemic and related health measures on trauma admissions. The consistency of this 11-week period across both years mitigates seasonal injury pattern variations. The criteria included cases of head trauma (ICD-10 codes S01–S09), neck trauma (S09–S19), thoracic trauma (S21–S29), and abdominal trauma (S31–S39), with hospitalizations during the specified 11-week period in both years. Exclusions encompassed records lacking vital data, limb traumas, and patients aged 0 or below. The exclusion of limb traumas focused the study on more severe injuries, which have greater implications for hospitalization and morbidity. The initial dataset comprised 17,776 records from 2019 and 11,407 from 2020.

### 2.2. Participants

The study analyzed complete datasets from two periods: 2018–2019 and 2020–2021. The data included essential details such as sex, place of residence, age, ICD-10 disease code, and admission date for patients treated for trauma in 35 hospitals across the Lublin Province. These datasets were obtained from the NGHMS and were rigorously vetted to meet the criteria for inclusion in the study. The inclusion criteria encompassed a broad range of trauma cases, providing a representative overview of the patient population affected by traumatic injuries in the region.

### 2.3. Ethical Considerations

As a descriptive study examining retrospective data, our research was exempt from requiring approval from a research ethics committee in Poland, following national law and ethical guidelines. The study did not involve collecting human biological samples or accessing personal or sensitive information. Compliance with regulatory standards was maintained in accordance with the Medical and Dentist Profession Act of 5 December 1996, the Human Clinical Act of 9 March 2023, and the Regulation of the Minister of Health of 26 January 2023 on Bioethical Commissions.

### 2.4. Statistical Analysis

Continuous data were presented as mean ± standard deviation (SD) or median with interquartile range (IQR). Categorical data were expressed as numbers and percentages. The Chi-squared test was utilized for the comparison of categorical data, while the Student’s *t*-test was employed to compare means between categories. These statistical tests were chosen for their suitability in analyzing continuous and categorical data, respectively. Additionally, logistic regression was used to assess the relationships between trauma types and patient demographics, including age, sex, and place of residence. The results of the logistic regression are presented as beta coefficients with 95% confidence intervals (CI). Separate multivariable models were developed to include trauma types with age and trauma type with place of residence as variables. Statistical analyses were conducted using PSPP Statistics for Mac, Version 8. A *p*-value of less than 0.05 was considered statistically significant.

## 3. Results

### 3.1. Participant Demographics

This study encompassed data from two periods: 17,773 patients for 2018–2019 and 11,394 for 2020–2021. Our analysis revealed significant demographic trends within these cohorts. The majority of patients in both periods were male, approximately 70%, with no significant statistical variance between the two timeframes. Notably, the average age differed significantly between the cohorts; it was lower in 2018–2019 (Mean = 44.30, SD = 24.16) compared to 2020–2021 (Mean = 47.70, SD = 23.47), with a *p*-value of <0.001. The place of residence also shifted over these periods, moving from a rural majority (54%) in the first cohort to urban predominance (52%) in the latter. Table 1 succinctly summarizes these demographic details, presenting age, gender distribution, and residence, alongside their statistical significance.

### 3.2. Comparative Characteristics of Study Cohorts

Our analysis of 17,773 patients from 2018–2019 and 11,394 from 2020–2021 highlighted key demographic differences. Across both cohorts, males comprised around 70% of the patients. A notable shift was observed in patient ages, with the mean age being significantly lower in the 2018–2019 cohort (M = 44.30) compared to 2020–2021 (M = 47.70, *p* < 0.001). The place of residence also varied significantly, shifting from a rural majority in the earlier period (53%) to an urban majority in the latter (52%).

### 3.3. Admittances

Our study revealed a significant decrease in trauma admissions during the pandemic (2020–2021) compared to the pre-pandemic period (2018–2019). Before the pandemic, there were 17,981 admissions, while during the pandemic, a marked increase in female admissions for abdominal injuries was noted, rising from 43% to 53%.

Head injuries (S01–S09) were the predominant cause of admissions, accounting for 69% in 2018–2019, decreasing to 63% in 2020–2021. This reduction was statistically significant. Table 2 provides a summarized comparison of trauma types based on urban and rural residency over these periods.

### 3.4. The Period of 2018–2019

In 2018–2019, there was a statistically significant rise in trauma admissions compared to 2020–2021. Rural residents accounted for a larger portion of admissions (54%). Head injuries dominated with 12,384 cases, mostly among male patients (73%) and notably from rural areas (53%). Neck injuries were less frequent but followed a similar pattern in gender and rural-urban distribution. Chest injuries (2038 cases) and abdominal injuries (2176 cases) also showed higher occurrences in rural settings and among male patients.

### 3.5. The Period of 2020–2021

During the pandemic years, there was a notable decline in trauma admissions. Head injuries still led, constituting about half of all cases, predominantly affecting male patients. Neck injuries decreased, with a slight urban majority, while chest injuries were more common in rural areas. Interestingly, a shift in abdominal injuries was observed, with a majority of urban cases being female, contrasting with the male majority in rural areas.

### 3.6. Fatalities Average Length of Hospitalization in Rural Areas

In the 2018–2019 period, there were 326 fatalities due to head injuries, with females constituting 30% of these cases. This gender distribution in fatalities was statistically significant. Neck injuries resulted in 11 deaths (27% female), chest injuries in 29 deaths (34% female), and abdominal injuries in 35 fatal cases (31% female), showing a consistent trend of male predominance in fatalities.

In contrast, the 2020–2021 period saw an increase in fatalities from head injuries, totaling 465 cases, with females representing 36%. Neck injuries were fatal in 18 instances (22% female), chest injuries in 28 (36% female), and abdominal injuries in 37 cases (43% female). The increase in female fatalities for head injuries during this period was notable, maintaining the overall male predominance trend.

Table 3 presents a detailed summary of fatalities according to the type of trauma across these periods.

### 3.7. Fatalities

The average hospital stay for patients from rural areas showed variation based on the type of injury and the period of analysis. Notably, for head injuries, the average hospitalization duration increased from 3.5 days in 2018–2019 to 5.5 days in 2020–2021, with males experiencing longer stays than females in both periods. Similarly, neck injury cases saw an increase in average hospitalization from 4 to 5 days over the same timeframes, again with males having longer stays. In the case of chest injuries, the average hospital stay slightly rose from 6 days in 2018–2019 to 6.5 days in 2020–2021. The earlier period saw equal hospitalization times for both genders, but in the latter period, females had longer hospital stays. For abdominal injuries, there was a decrease in average hospital stay from 7.5 days in 2018–2019 to 6.5 days in 2020–2021, with males typically hospitalized longer than females. Table 4 outlines the hospitalization lengths by injury type and gender in rural areas.

An important aspect observed in our data is the discrepancy in the number of fatalities relative to the total number of patients between the two periods. Despite the decline in trauma admissions during 2020–2021 compared to 2018–2019, there was an increase in fatalities in the later period. Specifically, fatalities rose from 434 in 2018–2019 to 634 in 2020–2021, despite a reduction in overall patient admissions from 17,773 to 11,394. This trend suggests a significant impact of the pandemic on trauma outcomes.

### 3.8. Average Length of Hospitalization and Maximum Hospitalization Days

The analysis of hospitalization durations revealed that urban areas saw an increase in the average hospitalization duration for head injuries from 3.5 days in 2018–2019 to 4 days in 2020–2021. This rise was more pronounced in rural areas, where it increased from 3.5 days to 5.5 days over the same period. Additionally, the maximum length of stay for head injuries also increased, with urban areas recording an average of 184 days in 2020–2021, up from 166 days, and rural areas seeing an increase to 135 days from 134 days. In both urban and rural settings, male patients generally experienced longer hospitalizations and maximum stay durations.

The average hospital stay for neck injuries in urban areas also saw a rise, increasing from 2.5 days to 3.5 days. The maximum duration for these injuries notably increased from 90 days in 2018–2019 to 152 days in 2020–2021. In rural areas, the maximum hospitalization duration rose from 178 days to 196 days. Similar to head injuries, male patients had longer stays for neck injuries.

For thoracic injuries, the average hospitalization duration in urban areas slightly increased from 5 days to 6 days, while rural patients experienced an increase from 6 days to 6.5 days. Interestingly, the maximum length of stay for thoracic injuries showed a decrease in both urban and rural areas.

In the case of abdominal injuries, there was a decrease in the average hospital stay in both urban and rural areas, dropping from 7 days to 6.5 days and from 7.5 days to 6.5 days, respectively. The maximum stay duration also decreased for these injuries in both demographics.

Overall, the data suggests a consistent pattern of longer hospitalizations for male patients and reveals significant changes in hospitalization durations based on the type of injury and geographical location. Particularly in the 2020–2021 period, patients in rural areas experienced longer maximum hospitalization days for head and neck injuries compared to their urban counterparts. Conversely, in the 2018–2019 period, rural patients with chest and abdominal injuries had longer maximum hospital stays than urban patients. These disparities highlight the impact of geographical location on the trajectory of healthcare for trauma patients. Table 4 provides a detailed breakdown of hospitalization lengths by injury type and gender.

### 3.9. Relationship between the Type of Trauma and Selected Variables in Logistic Regression Analysis

In our study, we employed logistic regression analysis to explore the impact of so-ciodemographic factors, specifically age and territorial postcode, on different types of trauma. This analysis incorporated data from both the 2018–2019 and 2020–2021 periods, providing a comprehensive view across these timeframes. The aim was to assess whether age and geographical area, represented by territorial postcode, significantly influenced the incidence and nature of trauma cases.

As detailed in Table 5, the analysis revealed that age is a significant predictor of trauma type. The regression coefficient for age was stable and non-zero, indicating a robust relationship. This finding suggests that age plays a crucial role in both the likelihood and nature of trauma experienced by individuals. We further augmented our analysis by calculating the Odds Ratios (OR) and their 95% Confidence Intervals for each variable. The OR provides a measure of association between an exposure and an outcome, and the CI indicates the precision of the OR estimate.

In contrast, the influence of geographical location, as represented by the territorial postcode, was minimal. This was evidenced by a very low coefficient and standard error, and a corresponding OR close to 1, indicating that the geographical location, at least as defined by postal codes, may not be as impactful in determining the nature of trauma experienced by patients.

## 4. Discussion

In our descriptive analysis, we observed a discernible decline in trauma admissions during the pandemic years (2020–2021) across both urban and rural settings. This aligns with global trends, where a reduction in trauma cases has been noted as an indirect effect of COVID-19-related restrictions, which significantly altered daily activities and mobility [33]. An increase in the average length of hospital stays for trauma patients was also recorded during the pandemic, more prominently in rural areas. This finding mirrors global reports where healthcare systems’ responsiveness was challenged, leading to longer hospitalization durations due to altered triage and care protocols [34].

It is important to underscore that our study’s findings are primarily descriptive, reflecting trauma admission trends in specific timeframes without inferring causality or strong associations. The observed changes in trauma admissions and hospitalization patterns are presented as they occurred in our data set, without adjustments for potential confounding factors. Therefore, while our results provide valuable insights into trauma care dynamics during the pandemic, they should be interpreted as descriptive observations rather than definitive conclusions about cause-and-effect relationships.

An intriguing observation from our study was the discrepancy in the number of fatalities relative to the total number of patients, particularly during the pandemic years (2020–2021). Despite a decrease in overall trauma admissions during this period, there was an unexpected increase in the total number of fatalities. This trend raises critical concerns about the severity and complexity of trauma cases during the pandemic. It potentially indicates that while there were fewer trauma incidents, those that occurred may have been more severe, or that there were challenges in providing timely and effective trauma care during the pandemic. This aspect highlights the importance of not only focusing on admission rates but also on the quality and outcomes of care provided to trauma patients, especially during public health crises.

The extended average length of hospital stays, especially pronounced in rural areas, raises questions about the capacity of healthcare systems to respond to surges in demand during crises. The disparity between urban and rural healthcare experiences may indicate underlying systemic issues that extend beyond the immediate effects of the pandemic, such as healthcare staffing, infrastructure disparities, and the efficiency of care pathways. Addressing these disparities is crucial for reducing the urban-rural healthcare divide and ensuring that all patients have access to timely and effective trauma care.

Our study not only documents a reduction in trauma admissions during the COVID-19 pandemic but also highlights an interesting shift in the demographic profile of trauma patients. The relative increase in trauma admissions among women for abdominal injuries may reflect changes in the societal dynamics and risk exposure during the pandemic, underscoring the influence of social behavior changes on injury patterns [35]. This rise could be attributed to the altered societal dynamics during the lockdown periods. The increase in time spent at home, changes in daily routines, and perhaps a heightened involvement in household activities, including cooking and childcare, could have contributed to this trend. Moreover, the pandemic’s impact on mental health and stress levels might have played a role in altering the risk patterns for injuries. These observations highlight the complex interplay between societal changes and injury patterns, underscoring the need for context-specific public health strategies [36].

The preponderance of trauma cases in males persisted as a consistent theme across both periods, with a notable uptick in severe injuries and fatalities compared to females. This gender-related disparity is corroborated by existing literature that often attributes it to higher-risk behaviors and occupational exposures typically associated with males [37]. However, the pandemic period’s unexpected rise in female admissions for abdominal injuries suggests a potential shift in injury patterns, possibly due to the increased domestic workload and activities during lockdowns [38].

Our analysis sheds light on the disparities in the maximal hospitalization days between urban and rural patients, with the latter experiencing longer stays. This discrepancy could reflect delayed access to care, varying degrees of injury severity upon presentation, or differential resource allocation that often favors urban healthcare facilities [39]. The pandemic’s impact, which saw an extension in hospital stays for head and neck injuries in rural settings, could suggest a resource diversion phenomenon, where urban healthcare systems, being at the forefront of the pandemic response, might have facilitated quicker discharges to accommodate the influx of COVID-19 patients [40].

The extended average length of hospital stays, especially pronounced in rural areas, raises questions about the capacity of healthcare systems to respond to surges in demand during crises. The disparity between urban and rural healthcare experiences may indicate underlying systemic issues that extend beyond the immediate effects of the pandemic, such as healthcare staffing, infrastructure disparities, and the efficiency of care pathways [41]. Addressing these disparities is crucial for reducing the urban-rural healthcare divide and ensuring that all patients have access to timely and effective trauma care [42].

The study’s findings are a clarion call for health systems to enhance their adaptability and resilience, especially in rural areas where disparities in healthcare access and efficiency became more pronounced during the pandemic [43,44]. The increased hospitalization lengths during the pandemic also prompt a critical evaluation of healthcare processes, aiming to balance care quality with operational efficiency.

Through this research, we contribute to the nuanced understanding of how trauma care delivery is affected by patient demographics, geographic factors, and larger-scale public health crises like pandemics. These insights emphasize the importance of agile public health interventions and equitable resource distribution to maintain the quality of trauma care during times of widespread healthcare disruption.

Furthermore, our findings regarding the maximal number of hospitalization days, particularly for patients with head and neck injuries in rural areas, call attention to the need for enhanced trauma care systems that are not only robust but also flexible enough to adjust to sudden changes in healthcare demands [45,46,47]. The sustained or increased length of stay for trauma patients during the pandemic highlights a resilience gap in the healthcare system, which requires strategic investments in rural healthcare infrastructure to improve response capacity in future crises.

Lastly, our study emphasizes the importance of continuity in trauma care services, even as healthcare systems pivot to address pandemic-related challenges. The observed trends serve as a reminder that healthcare planning must be inclusive, accounting for the full spectrum of patient needs, including those unrelated to the immediate crisis, to prevent secondary health crises arising from neglected routine care.

Given the purely descriptive nature of our study, it is crucial to acknowledge the limitations inherent in this approach. Our analysis does not account for potential confounding factors that might influence trauma admission rates and hospitalization patterns. As such, our findings should be viewed as an initial step in understanding the impact of the COVID-19 pandemic on trauma care, offering a basis for future studies that might explore these relationships in greater depth using more complex statistical methods. Our study, therefore, serves as a descriptive snapshot of a unique period in healthcare, providing a foundation for more comprehensive, analytical research in the future.

These observations, it becomes evident that the COVID-19 pandemic has provided a unique lens through which the vulnerabilities and strengths of healthcare systems have been magnified. It offers a pivotal learning opportunity to reimagine and reinforce trauma care pathways, ensuring they are equitable, efficient, and resilient in the face of any public health emergency.

## 5. Limitations

Our descriptive analysis focused on recording trauma admission trends, carries inherent limitations, particularly in data accuracy and completeness. Biases in record-keeping and potential gaps in capturing all relevant patient or contextual information must be considered when interpreting our findings.

Geographically, our study is confined to the Lublin Province in Poland, limiting the generalizability of our results to other regions. Different healthcare systems, population behaviors, and varying impacts of the COVID-19 pandemic might render our findings less applicable elsewhere.

Our analysis also faced limitations in the granularity of available data. Important factors like injury severity, specific treatment protocols, and underlying health conditions, which could influence hospitalization durations and outcomes, were not available for our analysis.

The unique challenges posed by the COVID-19 pandemic, including strains on healthcare resources, staffing changes, and alterations in patient behaviors, might have also influenced our findings. These factors, coupled with the pandemic’s broader impacts, need to be factored into the interpretation of our results.

Our focus was primarily on the geographic location of patients (rural vs. urban) and did not encompass other demographic variables like socioeconomic status, occupation, or lifestyle, which could significantly impact trauma incidence and outcomes.

The selected study periods, 2018–2019 and 2020–2021, were intended to reflect the pre-pandemic and pandemic scenarios but may not represent longer-term trends in trauma care.

Finally, the potential for unmeasured confounders in our observational study cannot be discounted. While we attempted to account for known variables, other unknown factors could have influenced our findings.

Recognizing these limitations is essential, highlighting the need for further research. Future studies, particularly prospective ones incorporating a broader range of variables, are necessary to deepen our understanding of trauma care dynamics in the context of public health events like the COVID-19 pandemic.

## 6. Conclusions

Our analysis of hospitalization data from 2018 to 2021 reveals distinct trends in trauma patient hospital stays, highlighting disparities between rural and urban areas. In 2018–2019, rural patients typically had longer hospitalizations for neck, chest, and abdominal injuries, while during the pandemic years of 2020–2021, this trend persisted with some shifts in patterns, such as shorter stays for abdominal injuries in rural areas compared to urban counterparts. These shifts suggest the impact of healthcare system adaptations and changing injury profiles due to the pandemic.

This study underscores the need for tailored healthcare strategies to address rural-urban disparities, especially in response to crises like the COVID-19 pandemic. It emphasizes the importance of building a resilient healthcare infrastructure that offers equitable care across different geographical locations.

We recognize that our study is descriptive and does not establish causality. The observed differences should be viewed as indicative trends for further investigation. The disparities we noted point to systemic issues in trauma care delivery that necessitate further research and targeted policy interventions.

In light of our findings, we recommend healthcare providers and policymakers focus on developing adaptable healthcare systems, particularly in rural areas, to better manage future pandemics. This includes investments in infrastructure, equitable resource allocation, and enhanced training for healthcare personnel. Policy interventions should also aim at developing flexible public health strategies to minimize the impact of pandemics on trauma care.

In conclusion, our analysis provides a foundation for further research into trauma care dynamics influenced by geographic, societal, and healthcare system factors. Future studies with more analytical approaches are essential to deepen our understanding of these dynamics and inform strategies for equitable and effective trauma care, irrespective of geographical location or prevailing health challenges.

## Figures and Tables

**Table 1 jcm-13-00259-t001:** Characteristic of the research group.

Variables	2018–2019	2020–2021	*p*-Value
Age [year]			*p* ≤ 0.001 **
N	17,776	11,407	
M	44.30	47.70	
SD	24.16	23.47	
Gender			*p* = 0.082 *
Male	12,343 (69%)	7819 (68%)	
Famale	5433 (31%)	3588 (32%)	
Place of Residence			*p* ≤ 0.001
Urban Area	8242 (46%)	5482 (48%)	
Rural Area	9534 (54%)	5925(52%)	

* Significance level for the chi-squared test (χ²). ** Significance level for the Student’s *t*-test.

**Table 2 jcm-13-00259-t002:** Types of Trauma Divided into Urban and Rural Area Inhabitants.

ICD 10	Trauma Type	Urban Area Patients 2018–2019	Rural Area Patients 2018–2019	Urban Area Patients 2020–2021	Rural Area Patients 2020–2021
S01–S09	Head Trauma	5728	6444	3535	3747
S011–S019	Neck Trauma	718	713	445	443
S021–S029	Thoracic Trauma	813	1206	727	872
S031–S039	Abdominal Trauma	983	1171	775	863

Legend: head trauma S01–S09; neck trauma S09–S19; thoracic trauma S21–S29; abdominal trauma S31–S39.

**Table 3 jcm-13-00259-t003:** Fatalities According to the Type of Trauma in Analyzed Periods.

Type of Trauma	2018	2019	2020	2021
S01–S09 (Head Trauma)	166	160	181	172
S011–S019 (Neck Trauma)	5	6	3	13
S21–S29 (Thoracic Trauma)	16	13	16	12
S31–S39 (Abdominal Trauma)	16	19	20	17
Total	203	198	220	214

**Table 4 jcm-13-00259-t004:** Hospitalization Length by Injury Type and Gender in Rural Area.

Types of Trauma	Admitted Female Patients	Admitted Male Patients	Max Hospitalization Length in Female	Max Hospitalization Length in Male	Mean Hospitalization Length in Female	Mean Hospitalization Length in Male
Rural Areas 2018–2019						
S01–S09	1639	4805	98	170	3.5	4
S011–S19	300	413	42	314	4	5
S21–S29	309	897	117	193	6	6.5
S31–S39	456	715	110	251	7.5	8
Total	2704	6830				
Rural Areas 2020–2021						
S01–S09	900	2847	111	159	5.5	5
S011-S19	187	256	85	111	5	7
S21–S29	224	648	84	160	7	6
S31–S39	323	540	50	67	6.5	7
Total	1634	4291				

Legend: head trauma S01–S09; neck trauma S09–S19; thoracic trauma S21–S29; abdominal trauma S31–S39.

**Table 5 jcm-13-00259-t005:** Logistic Regression Analysis.

Variables	b (SE)	OR (95% CI)	*p*
Type of TRAUMA:			
Place of leaving:			
Teritorial postcode	2.50 × 10^−6^	1.00 (approx.) per unit increase	0.025
Type of TRAUMA:			
Age:	1.06	2.886 (2.718–3.065)	<0.001

b (SE): The regression coefficients (b) and their standard errors (SE); OR (Odds Ratio): The exponentiation of the regression coefficient, indicating the effect size and direction; 95% CI: The 95% Confidence Interval for the Odds Ratio, providing a range within which the true effect size is likely to fall.

## Data Availability

The datasets used and/or analyzed during the current study are available from the corresponding author on reasonable request.

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
