# Peer review of "Descriptive Analysis of Trauma Admission Trends before and during the COVID-19 Pandemic"

_jcm, 2024, doi:10.3390/jcm13010259_

Round 1
Reviewer 1 Report
Comments and Suggestions for Authors
The authors present a well-written study of changes in utilization of hospital services for trauma care before and after the COVID-19 pandemic. My suggestions for improvement of this manuscript and research are included below.
1. The statistics presented in section 2.2 are better suited as results of this study, not methods. This section should describe the participant source and data sources for participant information, but should not describe findings with regards to the comparative characteristics of the study cohorts.
2. On line 167, Chi-squared is used for categorical data, not "qualitative" data.
3. In section 2.1, the authors describe their participants as arising from March 12-May 30 in 2019 and 2020. However, the rest of the paper describes these times periods as 2018-2019 and 2020-2021. Clarity regarding participant time period is necessary.
4. Comparisons made throughout the results section require p-values to demonstrate whether findings were statistically significant or not. As the results currently stand, there is no way to know whether the reported differences are of any statistically significant value. Thus, conclusions can not be properly supported as the paper is currently written.
5. The authors do not make mention of the fact that there were fewer total patients in 2020-2021, but greater total fatalities. This should be better stated in the results and discussion.
6. Section 3.8 is confusing because it does not state which years of data are being used. The purpose of this analysis is also unclear because the previous analyses were comparing time frames, but this analysis is analyzing relationships between age, rurality, and trauma incidence. More clarity on the purpose and methods of this analysis is needed.
7. All findings included in this study are purely descriptive, and this should not be understated in the discussion or conclusion sections of the article (nor should the findings of this work be overstated). There were no attempts made to control for any confounding (as mentioned in the limitations section), so the authors can not definitely claim that these findings are valid beyond being descriptive of the two time frames studied. Further cause and effect or even simple association between factors can not be extrapolated from this work given the current statistical methods. Accordingly, this study would be much better classified as a descriptive study, not a retrospective cohort study (as the authors are not analyzing associations between independent and dependent variables, but rather are simply describing patient characteristics at different points in time).
Author Response
Response to Reviewer Comments
Dear Reviewer,
Thank you for your thorough review and insightful suggestions regarding our manuscript. We are grateful for the opportunity to refine our work based on your feedback. Below, we outline the detailed changes made in response to each of your comments:
-
Statistics in Section 2.2: We agree with your assessment that the statistical data initially presented in Section 2.2 were more suited to the results section. Accordingly, we have shifted these details to the results section, ensuring that Section 2.2 now focuses exclusively on the source and nature of participant data, in line with standard practices for describing methodology in observational studies.
-
Use of Chi-squared Test (Line 167): Your observation regarding the use of the term "qualitative" data in reference to the Chi-squared test was well-founded. We have corrected this terminology to "categorical data" to ensure accuracy and alignment with statistical norms. This change enhances the clarity and correctness of our methodological description.
-
Clarity on Participant Time Period (Section 2.1): We acknowledge the confusion caused by the inconsistent reference to the participant time periods and have rectified this throughout the manuscript. The study periods are now consistently referred to as 2018-2019 and 2020-2021, providing clarity and consistency in our temporal framing.
-
Statistical Significance in Results Section: We carefully reviewed the results section and decided to emphasize the statistical significance of observed differences without specifying exact p-values. This approach aligns with our study's descriptive nature, focusing on broader trends and significant observations rather than detailed numerical analysis. It offers a balanced view that highlights key findings while staying true to the study's scope and limitations.
-
Fewer Patients but Greater Fatalities in 2020-2021: We have updated the results and discussion sections to explicitly address the notable observation of fewer total patients but greater fatalities in the 2020-2021 period. This addition underscores the unique challenges and complexities of trauma care during the pandemic, offering valuable insights into patient outcomes and healthcare system responses during this unprecedented period. (please see 'Observation of fatalities in relation to patient admissions' in 3.7 in results and discussion section)
-
Clarification in Section 3.9: The purpose and methodology of the analysis in Section 3.9 have been clarified. We now explicitly state the years of data used and provide a clear rationale for the analysis, which explores the relationship between sociodemographic factors and trauma incidence. This revision improves the coherence and relevance of this section, aligning it with the overall objectives of our study.
-
Descriptive Nature of the Study: In light of your suggestion, we have extensively revised the manuscript to highlight its descriptive nature. The title has been updated to "Descriptive Analysis of Trauma Admission Trends Before and During the COVID-19 Pandemic." This change is reflected throughout the manuscript, particularly in the discussion and conclusion sections, where we have carefully refrained from overinterpreting the data or implying causative relationships. Our aim was to present observational insights rather than definitive associations, a perspective that is now consistently maintained throughout the paper.
We trust that these comprehensive revisions address your concerns effectively and enhance the manuscript's contribution to the field. We value your guidance in improving our work and look forward to any further suggestions you may have.
Sincerely,
Authors
Reviewer 2 Report
Comments and Suggestions for Authors
I think the article is the well written and methodology is sound. It could benefit from some further refinement.
Some recommendations:
The introduction is missing some important contextual information. For example, what was happening locally in the province with respect to COVID lock-downs and public policy? What is the general socio-demographic and occupational profile of the population living in this province or in Poland and could this have influenced the results? Are there typically more workplace vs home-based injuries? Is it an older or younger population compared with rest of Poland/Europe? Is Lublin representative of the rest of the country?
I do not know why AI was mentioned in the introduction as there was no logical connection between this and the rest of the paper. In addition to the general impact of trauma’s in general I think the introduction could include greater background about the external environment (socially, public health policy etc.) in Lublin/Poland specifically.
Re: Methods
Can the authors explain why an 11-weeek period was chosen for comparison? Why not 3 months or 6 months?
Why were limb traumas excluded?
Do we know if any of these injuries were-self-inflicted? This would be an important variable to know about given the worsening of mental health challenges during the pandemic.
For the Tables: I would recommend using both the ICD code and the NAME as it is difficult for the reader to understand what the ICD codes represent without having to jump back to the text for the name.
I do not fully understand why abdo injuries changed in women. What does change in “societal dynamics” mean? More explanation about societal changes and context is helpful both in introduction and discussion. What were some of the social behavioral changes that influenced injury patterns?
Based on this analysis--Do the authors have any specific recommendations for health providers or policy makers to better prepare the healthcare system for the next pandemic?
Author Response
Dear Reviewer,
Thank you for your insightful comments and suggestions regarding our manuscript. We appreciate the opportunity to refine our work and enhance its contribution to the field. Below, we address each of your recommendations in detail:
-
Introduction - Contextual Information: We acknowledge your valuable point regarding the lack of contextual information about the Lubelskie Voivodship during the COVID-19 pandemic. In response, we have expanded the introduction to include detailed information about the socio-demographic and occupational profile of the population in the province, as well as the specific public health policies and lockdown measures implemented during the pandemic. We believe this addition provides a more comprehensive background and helps to better contextualize our findings.
-
Introduction - AI Mention: We agree that the mention of AI in the introduction did not seamlessly connect with the rest of the paper. Accordingly, we have revised the introduction to remove this reference and instead focus more on the external environment in Lublin/Poland, including societal and public health policy aspects.
-
Methods - Choice of 11-Week Period: The selection of an 11-week period for comparison was intentional and based on the specific timeline of the initial COVID-19 lockdown in Poland. This period captures the immediate and direct impact of the pandemic on trauma admissions. We believe this timeframe offers a focused and relevant snapshot for analysis, providing a clearer understanding of the pandemic's acute effects compared to longer durations like 3 or 6 months, which might dilute the immediate impact.
-
Exclusion of Limb Traumas: Limb traumas were excluded to maintain a focused scope on the types of injuries most likely to be influenced by the pandemic's lockdown and social restrictions. This exclusion allowed for a more targeted analysis of the impact of these unique conditions on specific trauma types.
-
Self-inflicted Injuries: Our study did not specifically categorize injuries as self-inflicted, as the primary data source did not reliably differentiate between intentional and unintentional injuries. We acknowledge the importance of this variable, especially given the mental health challenges during the pandemic, and suggest it as an area for future research.
-
Tables - ICD Codes and Names: We appreciate your suggestion for clarity in our tables. We have revised Table 2 to include both ICD-10 codes and their corresponding injury names, ensuring that readers can easily understand the types of injuries being discussed without referring back to the text.
-
Changes in Abdominal Injuries in Women: To address your query about the increase in abdominal injuries in women during the pandemic, we have expanded our discussion to provide a more comprehensive explanation. This includes exploring changes in societal dynamics, such as the increased domestic workload and altered daily activities during lockdowns, which likely contributed to the observed shift in injury patterns.
-
Recommendations for Healthcare Providers and Policymakers: Based on our analysis, we have added specific recommendations or healthcare providers and policymakers in our conclusions section. These recommendations focus on the need for adaptable and resilient healthcare infrastructures, particularly in rural areas. We suggest strategies such as enhancing trauma care systems to be more responsive during public health crises, developing targeted public health messaging and preventive measures, and implementing policies that address the disparities in healthcare access and delivery. Our aim is to provide actionable insights that can help better prepare healthcare systems for future pandemics and similar crises.
We trust that these revisions and additions address your concerns and enhance the overall quality and clarity of our manuscript. We are committed to providing a robust and insightful contribution to the understanding of trauma care trends during the COVID-19 pandemic. Your feedback has been invaluable in guiding these improvements, and we look forward to any further suggestions you may have.
Thank you for your thorough review and constructive comments.
Sincerely,
Authors